Microplastics do not affect bleaching of Acropora cervicornis at ambient or elevated temperatures

Plafcan Martina M. mplafcan@usf.edu
Stallings Christopher D.
College of Marine Science, University of South Florida , St. Petersburg , FL , United States of America
de Rezende Carlos
Electronic publication date: 2022 Jun 17
Publication date: 2022
Volume: 10
Electronic Location ID: e13578
Received 2022 Jan 24; Accepted 2022 May 22
Copyright: ©2022 Plafcan and Stallings
Copyright year: 2022
Copyright holder: Plafcan and Stallings
License: This is an open access article distributed under the terms of the Creative Commons Attribution License, which permits unrestricted use, distribution, reproduction and adaptation in any medium and for any purpose provided that it is properly attributed. For attribution, the original author(s), title, publication source (PeerJ) and either DOI or URL of the article must be cited.
License URL: https://creativecommons.org/licenses/by/4.0/

Keywords: Ocean warming, Acroporid, Thermal stress, Microbeads, Laboratory experiment, Plastic pollution, Coral symbionts, Coral-reef ecosystems, Heterotrophic coral, Synergistic stressors

Funding: The Anne & Werner Von Rosenstiel Fellowship The Linton Tibbetts Endowed Graduate Student Fellowship The Fish Florida Scholarship The Gumbo Limbo Research Grant The University of South Florida’s College of Marine Science This work was funded by the Anne & Werner Von Rosenstiel Fellowship, the Linton Tibbetts Endowed Graduate Student Fellowship, the Fish Florida Scholarship, the Gumbo Limbo Research Grant, and the University of South Florida’s College of Marine Science. The funders had no role in study design, data collection and analysis, decision to publish, or preparation of the manuscript.

==============================
Microplastic pollution can harm organisms and ecosystems such as coral reefs. Corals are important habitat-forming organisms that are sensitive to environmental conditions and have been declining due to stressors associated with climate change. Despite their ecological importance, it is unclear how corals may be affected by microplastics or if there are synergistic effects with rising ocean temperatures. To address this research gap, we experimentally examined the combined effects of environmentally relevant microplastic concentrations (i.e., the global average) and elevated temperatures on bleaching of the threatened Caribbean coral, Acropora cervicornis. In a controlled laboratory setting, we exposed coral fragments to orthogonally crossed treatment levels of low-density polyethylene microplastic beads (0 and 11.8 particles L−1) and water temperatures (ambient at 28 °C and elevated at 32 °C). Zooxanthellae densities were quantified after the 17-day experiment to measure the bleaching response. Regardless of microplastic treatment level, corals in the elevated temperature treatment were visibly bleached and necrotic (i.e., significant negative effect on zooxanthellae density) while those exposed to ambient temperature remained healthy. Thus, our study successfully elicited the expected bleaching response to a high-water temperature. However, we did not observe significant effects of microplastics at either individual (ambient temperature) or combined levels (elevated temperature). Although elevated temperatures remain a larger threat to corals, responses to microplastics are complex and may vary based on focal organisms or on plastic conditions (e.g., concentration, size, shape). Our findings add to a small but growing body of research on the effects of microplastics on corals, but further work is warranted in this emerging field to fully understand how sensitive ecosystems are affected by this pollutant.

Introduction

Coral reefs provide recreational, commercial, and ecological services, which makes them a valuable marine habitat (Woodhead et al., 2019). Despite their importance, coral reefs are threatened by a suite of global and local stressors. Globally, climate change is affecting ocean temperatures which are expected to increase by 2.6–4.8 °C at the surface by 2100 (Pachauri et al., 2014; Rogelj, Meinshausen & Knutti, 2012), and can result in coral bleaching. The global effects of ocean warming on coral reefs are evidenced by the significant degradation and collapse of reef ecosystems since bleaching can lead to coral mortality (Pratchett et al., 2018). Due to the continual rise in ocean temperatures, there has been an increase in the frequency and intensity of coral bleaching events (Hughes et al., 2018; Riegl et al., 2009). The Florida Keys and Caribbean are among the most degraded reefs, with a 63% continuous decline in coral cover between 2007 and 2016 (Jones, Figueiredo & Gilliam, 2020), however reef degradation began decades before the recent changes (Schutte, Selig & Bruno, 2010). In addition to rising water temperatures, there is a growing concern about the effects of microplastics on coral-reef systems. Although some early studies have demonstrated that microplastics can negatively harm corals (Hankins, Duffy & Drisco, 2018; Reichert et al., 2018; Tang et al., 2018; Tang et al., 2021), the responses have been equivocal among species examined (Reichert et al., 2019; Reichert et al., 2018). We therefore lack an understanding of how microplastics may interact with elevated ocean temperatures, and how this emerging stressor may affect bleaching in most coral species. Addressing this research gap will help us to broaden our understanding of the generalities of the individual and combined effects of these two anthropogenic stressors on sensitive coral-reef ecosystems.

Exposure to microplastics in corals has been demonstrated to cause a variety of negative effects. Adhesion of microplastics to a coral’s surface can cause localized bleaching and tissue necrosis (Reichert et al., 2018), but further harm can occur when corals ingest them. There is some evidence to suggest corals accidentally ingest microplastics when they try to capture food (Axworthy & Padilla-Gamino, 2019). This prevents corals from obtaining real food due to time spent handling the plastic (Savinelli et al., 2020) and imparts satiation by filling their gastrovascular cavity (Rotjan et al., 2019). These responses can have important implications on their energy budgets because the movements involved with capturing, ingesting, and egesting microplastics are energetically costly (Reichert et al., 2019). In addition, a reduction in food consumption to replenish energy lost when handling the microplastics could ultimately cause an energy deficit (Savinelli et al., 2020). This may have profound repercussions when corals are stressed, such as in ocean warming conditions, since they need energy to cope with these stressors. However, only three studies to date have examined how microplastics and ocean warming interact in corals. Reichert et al. (2021) found equivocal effects of microplastics on five species of coral. Although microplastics exacerbated the effects of temperature on bleaching in one species, it did not affect bleaching in three species, and even reduced it in one (Reichert et al., 2021). Increased photosynthetic efficiency, upregulation of heat shock proteins, or increased heterotrophic feeding were potential explanations for why Montipora digitata bleached less when thermally stressed (Reichert et al., 2021). However, Axworthy & Padilla-Gamino (2019) found corals reduced feeding on Artemia but not on microplastics following thermal stress and suggested this could cause an energy deficit. Additionally, Mendrik et al. (2021) observed reduced photosynthetic activity in Acropora spp. exposed to microplastic fibers at ambient temperature likely due to an increase in reactive oxygen signaling species, an indicator of stress, but this effect was not found at high temperatures. The authors suggested the corals acclimated to thermal stress by producing oxidative enzymes which also protected them from the microplastic stress (Mendrik et al., 2021). Ultimately, the stress and energy deficits caused by microplastics combined with stress from elevated temperatures could interact to produce either an additive or synergistic effect on coral bleaching, but further work is needed to examine this.

Acropora cervicornis is an important reef-building species in the tropical western Atlantic region that provides ecosystem services such as habitat for organisms and storm protection of shorelines (Moberg & Folke, 1999; Woodhead et al., 2019). This species is particularly susceptible to bleaching and other stressors and has been declining in abundance over time (Langdon et al., 2018). In fact, A. cervicornis has been listed as critically endangered by the International Union for Conservation of Nature (Aronson et al., 2008), and some estimates have suggested it may not survive past 2035 due to its susceptibility to bleaching (Langdon et al., 2018). Its recent decline in abundance, combined with fast growth rates, reliance on asexual propagation, and ecological importance have made it a focal species for restoration efforts in the Caribbean (Johnson et al., 2011; Young, Schopmeyer & Lirman, 2012). A. cervicornis has been shown to ingest microplastics (Hankins, Moso & Lasseigne, 2021) but the effects of doing so remain unclear. Given the prevalence of microplastics in Caribbean waters (Garces-Ordonez et al., 2021; Rose & Webber, 2019), the warming trend in the region (Chollett et al., 2012; Kuffner et al., 2014), and the drastic declines of A. cervicornis (Aronson et al., 2008), it is imperative to assess the effects of the combined stressors (microplastics and elevated temperatures) on this sensitive coral. To address this knowledge gap, we asked: Does microplastic exposure interact with elevated water temperatures to exacerbate bleaching in A. cervicornis? To test this study question, we performed controlled laboratory experiments where we manipulated temperature and microplastic concentrations and quantified the amount of bleaching or tissue loss.

Materials & Methods

We conducted experiments in the University of South Florida’s College of Marine Science (CMS) aquarium facility. Mote Marine Laboratory (Summerland Key, Florida, USA) donated A. cervicornis fragments comprising two genotypes from several colonies each; the genotypes had moderate to high tolerance to heat stress (Muller et al., 2021). Corals were obtained from Mote Marine Laboratory under National Marine Sanctuary Permit FKNMS-2015-163-A3. We performed the experiments on coral fragments of the moderately heat-tolerant genotype in November 2020 and the high heat-tolerant genotype May–June 2021. We glued coral fragments to ceramic tiles upon arrival at the CMS and fed the corals 2.5 g per 100 gallons of a dried zooplankton mix per manufacturer recommendations (Reef-roids, PolypLab). We stored the fragments in a 190 L acclimation tank at the CMS for two weeks at 28 °C prior to the experiments. Lighting consisted of T5 High Output fluorescent lights (two 440 nm wavelength and two 15,000 K bulbs in each fixture in each tank) with an 8:16 h (light:dark) photoperiod. We used this photoperiod due to mortality associated with longer light periods in preliminary experiments, but this photoperiod is consistent with previous studies on corals maintained in the laboratory (Schutter et al., 2011). We used two submersible pumps (Model 3, Danner, Islandia, NY) to maintain circulation throughout the tank and a titanium heater to maintain temperature (Titanium 800+, Finnex) and controller (Apex Lite, Neptune Apex Systems). We made seawater with Reef Crystals Reef Salt (Instant Ocean, Blacksburg, VA) mixed with deionized water to a salinity of 35.

We used a fully orthogonal design to test the effects of temperature and microplastic exposure on coral bleaching. Specifically, we crossed two temperatures, 28 °C and 32 °C, with two microplastic concentrations, 0 microplastics L−1 and 11.8 microplastics L−1. We choose 28 °C to match the ambient water temperature at time of collection since Mote raised the corals in an offshore nursery. The higher temperature (32 °C) was within the predicted range for the tropical western Atlantic region by the year 2100 (Pachauri et al., 2014; Rogelj, Meinshausen & Knutti, 2012). The microplastic concentration reflected the global average of 11.8 microplastics L−1 (Barrows, Cathey & Petersen, 2018). We placed two-three coral fragments in each of the eight 8.26 L experimental tanks per treatment combination. We kept experimental tanks within a water bath to keep their temperature stable. Freytes-Ortiz & Stallings (2018) developed this system to examine the effects of ocean warming on marine organisms. We placed the heater in the water bath with pumps on opposite ends to circulate the water. Each experimental tank contained a wave maker (JVP-110 528 gallons hr−1, Sunsun, Zhoushan City, China) to generate flow and an airstone. We performed water changes of approximately one-third the tank volume every other day and measured water quality for eight parameters: temperature, calcium, alkalinity, nitrite, salinity, pH, nitrate, and ammonia. We randomly selected two tanks from each treatment four times throughout the experiment to test the water, and all tanks were ultimately examined. Water quality throughout each experiment was within an acceptable range except on the last day of the experiment for the moderately heat-tolerant genotype (Fig. 1). Two tanks had high levels of ammonia, nitrite, and nitrate caused by the tissue necrosis and mortality of the coral fragments in those tanks due to the elevated water temperature.

Figure 1 Water quality parameters (mean + SE) of tanks for each day measured throughout the experimental trials.

Circles are the ambient temperature tanks and triangles are the elevated temperature tanks. (A) Temperature. (B) Salinity. (C) Nitrite. (D) Nitrate. (E) Alkalinity. (F) Ammonia. (G) Calcium. (H) pH. Two tanks had high levels of ammonia, nitrite, and nitrate caused by tissue necrosis and mortality of the coral fragments in those tanks due to the elevated water temperature which raised those values for the last day of the experiment.

For the high temperature treatment, we increased the water temperature 0.5 °C each day until it reached 30 °C. We held the temperature at 30 °C for four days, then increased by 0.5 °C per day until it reached 32 °C where it remained constant for six days. This rate of temperature increase mitigated any effects of thermal shock. When the temperature was held at 32 °C, the tanks were maintained at 28 ± 0.02 °C (mean ± SE) and 32 ± 0.02 °C (mean ± SE). We added fluorescent green low-density polyethylene microbeads with a diameter range of 212–250 µm (1.025 g cc−1) and 300–355 µm (1.010 g cc−1) directly to the tanks at a concentration of 11.8 microplastics L−1 (5.9 particles L−1 of each size) (Barrows, Cathey & Petersen, 2018). We chose these microplastic sizes based on what the small-polyp A. cervicornis (1.26 mm−2.03 mm) can ingest. Prior to the experiments, we kept the microplastics in saltwater for at least one week to accumulate a biofilm. We added the microplastics to both the elevated and ambient temperature treatments after the first temperature increase along with food to initiate a feeding response. After the microplastics were added, they mostly floated on the surface on the first day and then were suspended in the water column for the remainder of the experiment. During water changes, we separated the microplastics and added them back to the tank to ensure consistent microplastic concentrations throughout the study duration. As a result of microplastic exposure, ingestion was an assumed response due to evidence by Hankins, Moso & Lasseigne (2021) and video we collected (Fig. 2, Video S1).

Figure 2 (A–B) Images of coral capturing a microplastic.

The black arrow in A points to the microplastic that was captured in B. Time stamps for each picture are at the bottom (note: these images were taken from Video S1 which is sped up 20x). Note that there is a second microplastic visible in A, but it was not captured by the coral during this recording. Both microplastics are circled in black in A.

We used a protocol to minimize contamination (Brander et al., 2020; Cowger et al., 2020), that we modified for corals. We separated the tanks from the rest of the room with a heavy-duty tarp to limit airborne contamination. We wore 100% cotton clothing to limit fiber shedding, thoroughly rinsed hardware (e.g., containers, glassware) with deionized water before use, and covered them in aluminum foil if not used immediately. We also rinsed our arms thoroughly with deionized water up to the elbows and wiped down all other surfaces with paper towels and deionized water.

To visually compare treatments throughout the experiments, we measured the response to thermal stress daily based on severity of coral bleaching and a visual estimate of percent surface area affected by tissue loss (i.e., necrosis). Coral bleaching occurs when the tissue loses its color due to the expulsion of zooxanthellae which makes the coral appear white, whereas tissue necrosis is the loss of tissue (Hoegh-Guldberg & Smith, 1989; Rodolfo-Metalpa et al., 2005). The ordinal bleaching scale we used was none (0), low (>0–25%), partial (25–50%), high (50–75%), and total (75–100%). Immediately following the conclusion of the experimental trials, we placed all corals in a −20 °C freezer for at least one hour, then removed them one at a time, and sprayed them with artificial seawater to remove the tissue (Johannes & Wiebe, 1970). We preserved collected tissue in 2% formalin. Next, we recorded the total homogenate volume (i.e., the volume of the zooxanthellae, seawater, and formalin), homogenized it, and counted zooxanthellae on 10 grids of a Neubauer-improved hemocytometer under a light microscope. To obtain the total zooxanthellae count for each fragment, we divided the average cell count per grid by the volume of the hemocytometer chamber, then multiplied by the total homogenate volume. We used the aluminum foil method from Marsh Jr (1970) to calculate the surface area of each fragment. To do this, we completely and snugly covered each coral skeleton in aluminum foil with no overlap, and then weighed the foil. Then we weighed five 100 cm2 foil sheets and calculated their mean mass as a reference. Next, we calculated the coral surface area by multiplying the reference foil surface area and coral foil weight then dividing by the reference foil weight. Finally, we quantified zooxanthellae density by dividing the zooxanthellae count of each coral fragment by its surface area.

To examine the additive and synergistic effects of temperature (fixed effect) and microplastics (fixed effect) on zooxanthellae density (response), we performed a generalized linear mixed model (GLMM) with tank included as a random effect. We determined the zooxanthellae response data were zero-inflated, and therefore examined several models that are capable of handling a large number of zeros (Zuur et al., 2009). We performed all analyses in R (R Development Core Team, 2021) using glmmTMB (Brooks et al., 2017) for the GLMM and DHARMa (Hartig & Hartig, 2021) for residual diagnostics. We used Akaike information criterion (AIC) to determine the best model then tested for diagnostics. We also determined that genotype did not affect zooxanthellae density (p = 0.55), and because we were not interested in its effects, per se, we pooled the data across genotypes. Our final model, that was deemed the best, was a zero-inflated, negative binomial model that examined the main effects of temperature and microplastic as well as an interaction between the two (AIC = 3,893.1).

Results

Bleaching did not occur in the ambient temperature (28 °C) treatment but was extensive in the elevated one (32 °C). Indeed, 97.5% of corals in the high temperature treatment were visibly bleached and 75.3% experienced tissue necrosis (Fig. 3). These observations held regardless of microplastic presence. Further, zooxanthellae density was strongly affected by elevated temperature (z = −8.15 p < 0.001, Table 1). However, zooxanthellae density was not affected by either microplastics alone (z = 1.07, p = 0.29) or in combination with elevated temperature (z = 1.04, p = 0.30). Neither elevated temperature (z = 0.01, p = 0.99) nor microplastic presence (z = 0.17, p = 0.87) contributed to excess zeros in the zero-inflated model.

Figure 3 Boxplot of zooxanthellae densities (100,000 cells * cm−2) for each treatment.

Presence of microplastics is indicated with MP- (absent) and MP+ (present). Temperature treatments, 28 °C and 32 °C, are indicated below the microplastic treatments. The boxes represent the median (horizontal line inside box), the first and third quartiles (lower and upper lines of the box, respectively) which shows the interquartile range, and the lower and upper whiskers represent the range within 1.5 * interquartile range. The additional point represents an outlier.

Discussion

Using a short-term laboratory experiment, we have shown that the presence of microplastics, when combined with thermal stress, did not alter the bleaching response of A. cervicornis. Importantly, these experiments were conducted using environmentally relevant microplastic concentrations. Research focused on the potential effects of microplastics on corals is an emerging field, and this study was one of the first to examine the orthogonal effects of microplastics with thermal stress (Axworthy & Padilla-Gamino, 2019; Reichert et al., 2021). As expected, elevated temperature reduced the zooxanthellae densities of the coral, but we found no individual or interactive effects of the microplastics.

The literature to date has been equivocal regarding the effects of microplastics on coral bleaching. The results from our study are consistent with previous research on Porites lutea and Heliopora coerulea at ambient temperature (Reichert et al., 2019; Reichert et al., 2018), but in contrast with studies that have found microplastic exposure can cause bleaching and tissue necrosis in A. muricata and Pocillopora verrucosa (Reichert et al., 2019; Reichert et al., 2018; Syakti et al., 2019). Similar to our study design, Reichert et al. (2021) examined the combined effects of microplastics and climate-change induced ocean warming, and found more severe bleaching in microplastic-treated fragments of Pocillopora verrucosa at elevated temperature. However, consistent with our results, Reichert et al. (2021) did not find an additive or synergistic effect of microplastics at elevated temperatures in A. muricata, Porites cylindrica, and Stylophora pistillata. The contrasting results among species highlights the species-specific responses corals have to microplastics.

Table 1 Output of GLMM to evaluate the effects of temperature and microplastic exposure on zooxanthellae density.

Ambient temperature (28 °C) and MP (absent) were used as model reference (α = 0.05, bold values indicate p < 0.05).

Conditional model	
	Coefficient	Std. Error	z value	Pr(>—z—)	
Intercept	13.944	0.057	245.27	<2e−16	
Temp32	−0.713	0.088	−8.15	3.69e−16	
MP2	0.085	0.080	1.07	0.287	
Temp32:MP2	0.128	0.123	1.04	0.298	
Zero-inflation model	
	Coefficient	Std. Error	z value	Pr(>—z—)	
Intercept	−19.698	2086.822	−0.009	0.992	
Temp32	18.850	2086.822	0.009	0.993	
MP2	0.080	0.481	0.166	0.868	
Random effects	
	Variance	Std. Dev.			
Tank	0.011	0.105			

Previous studies have attributed the different responses to microplastics among coral species to variation in their reliance on heterotrophic feeding (Reichert et al., 2019; Tang et al., 2021). Corals typically rely on photosynthesis to meet their energy demands but can supplement this with heterotrophic feeding (Grottoli, Rodrigues & Palardy, 2006), which makes them vulnerable to microplastics through ingestion. Microplastics have been shown to be stressful to corals (Tang et al., 2018), which can deplete their energy (Hankins, Moso & Lasseigne, 2021). In response to reduced energy, corals may increase heterotrophic feeding which leads to increased interactions with microplastics, additional stress, and energy depletion, subsequently causing bleaching (Reichert et al., 2019). Some coral species rely more on heterotrophic feeding than others, thus they are more vulnerable to microplastics while species that do not rely as much on heterotrophic feeding limit their interactions with microplastics and suffer less bleaching (Reichert et al., 2019; Reichert et al., 2018). This is especially concerning at elevated temperatures where corals can have heterotrophic plasticity in response to thermal stress (Grottoli, Rodrigues & Palardy, 2006), however we did not see an effect at either ambient or elevated temperatures. Microplastics were not stressful to A. cervicornis, possibly because they have small polyps that ingest less microplastics than large-polyp corals (Hankins, Duffy & Drisco, 2018; Hankins, Moso & Lasseigne, 2021). Despite a reliance on heterotrophic feeding (Towle, Enochs & Langdon, 2015), the smaller polyp size could have led to lower rates of microplastic ingestion which limited the interactions A. cervicornis had with the microplastics. Therefore, the stress and energy consumption associated with microplastic exposure was limited which prevented bleaching. However, it is unclear how many microplastics these corals ingested since the goal of this study was to assess the effects of microplastic exposure on coral bleaching rather than to specifically measure ingestion. Microplastic ingestion has been observed in this coral species, so we assumed it occurred throughout the experiments.

Experimental conditions may have also played a role in the lack of a microplastic effect in our study. For example, the response of corals to this pollutant has been shown to be dependent on microplastic concentration (Reichert et al., 2021; Syakti et al., 2019). The choice of concentration(s) to use in experimental studies can be complicated since they are dynamic both spatially (Barrows, Cathey & Petersen, 2018) and temporally (Courtene-Jones et al., 2020). Microplastic concentrations range from 0 to 220 particles L−1 in the global ocean (Barrows, Cathey & Petersen, 2018), 3 ×10−5 to 14 particles L−1 in the tropical western Atlantic Ocean, and approximately six particles L−1 in the Caribbean (Barrows, Cathey & Petersen, 2018; Ivardo Sul, Costa & Fillmann, 2014). Due to the large range of microplastic concentrations found in the global ocean, we used the global oceanic average to make it applicable to a broader range of locations. Our results align with previous work that did not find an effect of microplastics at concentrations reflective of current oceanic conditions (Bucci, Tulio & Rochman, 2020; Reichert et al., 2021; Syakti et al., 2019), whereas studies that have found stronger effects on zooxanthellae densities used 17 times, and higher, the concentration we used (Reichert et al., 2019; Reichert et al., 2018). For example, Reichert et al. (2021) found lower photosynthetic efficiency, mortality, and bleaching in two coral species when exposed to 2,500 microplastics L−1 at ambient and elevated temperatures but not at lower concentrations (2.5, 25, and 250 microplastics L−1). Our finding is important because it indicates that bleaching in A. cervicornis is not exacerbated by realistic microplastic concentrations observed on average in the global ocean, and ocean warming remains a larger threat. It is important to consider our experiments took place in a controlled laboratory setting and used a single, static microplastic concentration. However, corals can be exposed to temporally variable microplastic levels due to ocean dynamics which could result in a different response locally compared to a controlled laboratory setting. Microplastic size can also play an important role in the effects on organisms. For example, Syakti et al. (2019) found smaller microplastics had a stronger effect on bleaching compared to larger ones. Indeed, studies that assessed the effects of microplastics on corals have used a range of microplastic sizes from 1–500 µm, which could lend to the varying results. In this study, we used a mixture of two different microplastic sizes (212–250 and 300–355 µm) to simultaneously expose the corals to different sizes of plastic which is more representative of actual ocean conditions. In addition to microplastic concentration and size, the particle shape could have played a role in the lack of a response to the microplastics (Bucci, Tulio & Rochman, 2020; Mendrik et al., 2021). Photosynthesis in two coral species were altered in different directions (increase and decrease) by different microplastic shapes (fibers and spheres; Mendrik et al., 2021). Additionally, it remains unclear whether polymer type could affect responses to microplastics (Bucci, Tulio & Rochman, 2020). Indeed, most studies on corals, including ours, have used polyethylene microplastics (Axworthy & Padilla-Gamino, 2019; Hankins, Duffy & Drisco, 2018; Hankins, Moso & Lasseigne, 2021; Lanctôt et al., 2020). In contrast, few have used other polymer types (e.g., polystyrene, polypropylene) (Corona et al., 2020; Mendrik et al., 2021; Tang et al., 2018), so it is difficult to determine the role it may have on how corals respond to microplastics.

Conclusions

In our study, we orthogonally crossed temperature and microplastics to assess the effects of these combined stressors on bleaching in A. cervicornis. We found that microplastics had no effect on the bleaching response of A. cervicornis at ambient and elevated temperatures. Based on the minimal effect of microplastics observed in this study, A. cervicornis could be more tolerant to microplastics; however, further research will need to be conducted on this species to discern this. Also, our experiment assessed the short-term effects of microplastics combined with thermal stress on corals. Long-term experiments are needed to determine how organisms may respond to prolonged exposure to microplastics. While rising ocean temperatures remain a known major threat to corals, microplastic research on corals is still in its infancy. Future work should continue to test for the combined effects of microplastics and other stressors (e.g., ocean acidification, disease) in other coral species to understand how microplastics interact with previously identified stressors in coral-reef ecosystems. Additionally, studies should focus on using realistic microplastic concentrations to make their studies relevant to current and near future conditions but could also use a range of concentrations to identify whether response thresholds exist. Indeed, such efforts could be important since microplastic concentrations will likely continue to increase in the ocean as plastic production continues to grow. Such an effort would also add to the well-studied and often modeled effects of two other major anthropogenic stressors, global warming and ocean acidification.

Supplemental Information

Supplemental Information 1 Raw zooxanthellae density, bleaching, water quality, and temperature data

The zooxanthellae densities calculated for both genotypes. This data was used to determine if microplastics in combination with thermal stress exacerbated coral bleaching. Elevated temperature did have an effect on zooxanthellae density but microplastics did not. The water quality data show the water parameters measured on each day and the temperature data provide continuous measurements of the temperature in the tanks throughout the experiment. The bleaching data provide visual estimates of bleaching and necrosis to visually compare the different treatments throughout the experiments.

Click here for additional data file.

Supplemental Information 2 Full video of coral capturing a microplastic

This video is sped up 20×.

Click here for additional data file.

This project was made possible through the involvement of Mote Marine Laboratory and their collection of the corals used in this project. We thank Dr. Isabel Romero and Dr. Nicole Fogarty for their input and feedback on study design and early versions of the manuscript. We also thank Sharla Sugierski for her input in the study design for this work. We thank Ian Maier for his help transporting the corals and assisting throughout the experiments.

Additional Information and Declarations

Competing Interests

Author Contributions

Data Availability

The authors declare there are no competing interests.

Martina M. Plafcan conceived and designed the experiments, performed the experiments, analyzed the data, prepared figures and/or tables, authored or reviewed drafts of the article, and approved the final draft.

Christopher D. Stallings conceived and designed the experiments, analyzed the data, authored or reviewed drafts of the article, and approved the final draft.

The following information was supplied regarding data availability:

The raw zooxanthellae density data, daily bleaching data, water quality data, and temperature data are available in the Supplementary File.

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
