# Peer review of "Microplastics do not affect bleaching of Acropora cervicornis at ambient or elevated temperatures"

_PeerJ, doi:10.7717/peerj.13578_

## Round 0.1 · original submission · Minor Revisions

The study was reviewed by two researchers who have expertise in the area and I recommend that the authors make all the adjustments to be published in our journal. The topic is very important and every day we have new results that demonstrate the importance of comprehensive approaches to micro and nanoplastics dynamic in marine ecosystems as well as in marine organisms.

One of the reviewers requests that the experimental design be made clearer so that readers can properly understand the content of the study and I fully agree with this observation. One of the questions raised by a reviewer was exactly the difference between necrosis and bleaching, and this has to be clear, as there are fundamental differences between these two biological processes. There is a point that really needs to be addressed, which is whether or not microplastics are incorporated by organisms.

Finally, there is a recommendation to carry out a major revision in the writing of the manuscript to be more adequate in its final version, avoiding redundancies.

Reviewer 1 ·

Basic reporting

It is an interesting manuscript, a current thematic that considers the main disturbances on coral species: heating (bleaching) and marine debris (microplastics).
I suggest to review the discussion because the abstract, introduction and almost the entire discussion are all very similar. Reading one of these sections it is pratically the same. You need to bring new informations, raise new ideas, hypothesis on the discussion to improve it.
It will be very important to discuss about the potential of different types of microplastic, which probably should influence your results.

Experimental design

The experimental design did not consider the long-term effect of microplastic disturbance. This reflected in your results. Increasing temperature values (about 4 degrees) will have an immediate (few days) damage on corals, but increasing miscroplastic density will need longer period (probably months) to harm or impair the corals.

Validity of the findings

You need to show the artefact effect since your study was in a confined system and the ocean/sea is not. How could this interfere in your results.

Additional comments

What is the global oceanic average of microplastic concentrations? Please include these values and the references that mention it.

Annotated reviews are not available for download in order to protect the identity of reviewers who chose to remain anonymous.

Reviewer 2 ·

Basic reporting

The paper is written in very clear, unambiguous English throughout the text. I feel the literature is appropriately referenced, especially into the second paragraph of the introduction. Perhaps in paragraph #1, lines 49-50, when the authors talk about early studies, more than one reference (i.e., in addition to the Reichert et al. 2018) could be listed? Additionally, might some of the equivocal studies (line 51) be listed?

The overall structure is solid.

Regarding figures: On Figure 1 – does one want to mention the three panels that are significantly different (Nitrate, Nitrite & Ammonia?). Standing alone, a reader might recognize the potential, but I think having reference to that in the legend would be good.

Also, figure 2 – in color, it is appropriate. In black and white print, it is very difficult to see both the actual 2 microplastic beads and also the red arrow. Changing the arrow to black would solve the latter, perhaps adding small circles around the 2 beads on panel A would help with the second. There might be instances when the paper is desired to be printed, and B/W is more common, and cheaper.

Figure 3 = again, does one add statistics to the panel?

Table 1 – Why include the zero-inflation model? Especially if not referenced to in the text anywhere?

While the raw data is indeed, shared, I have some concerns with it, in diving deeper. First, it is somewhat difficult to follow: For example, it is never listed anywhere which tank is in which treatment. Digging deeper one can glean this information by sorting and organizing the data further, but it would seem helpful to perhaps maybe have 2 columns – one with TREAT TEMP (1 = 28; 2 = 32) and one with TREAT PLASTIC (1 = no plastic; 2 = plastic) or a column with four variables representing the 4 treatments: 1 = 28/no plastic; 2 = 28/plastic; 3 = 32/no plastic; 4 = 32/plastic. Additionally, (and this is a bit picky, I’ll admit), on the TEMP data page, the label TANK is not consistent with the TANKS on the other pages (i.e., there are 32 tanks), rather, on the TEMP page, Temp should likely be WATERBATH or some such.

Experimental design

Methods: It was not clear in the methods that the experiment was set up at two different times, one (11/2020) using the moderately heat tolerant corals and the other (6/2021) using the heat tolerant corals. This was quite confusing at first when looking at the raw data. Including the 2 time frames when the two different genotypes were done in the methods would be helpful.

Regarding bleaching data (paragraph starting line 165). It says bleaching and percent surface area affected by tissue loss were estimated DAILY. First, I take it “the area affected by tissue loss” in the text means “NECROSIS” on the data sheet? How exactly was necrosis calculated? Next: in looking at the raw data on bleaching tab in EXCEL, I don’t see daily estimates were made for all tanks, for example, from Tank 1’s in either the 2020 moderately heat or the 2021 data. Indeed, I don’t see 17 measurements (the length of the study from Fig. 1) for ANY tank in either coral genetic study. This implies the bleaching and necrosis data are NOT daily.

Finally, what is the exact difference in bleaching and necrosis? You don’t really define in the methods what they mean/relate to each other.

I would next start a new paragraph describing the Zoox Density data (i.e., after “IMMEDIATELY FOLLOWING on line 168). It might be helpful to provide a bit more background/ detail on some of the calculations here. For example, it would SEEM to me that the total volume of Zoo (total homogenate volume) would appear to be affected by the amount of seawater spray and the amount of formalin? Are either of those standard volumes? I’d like a bit more info. Also, as it appears that “foil weight” is important in the calculation of the coral surface area, more detail beyond referencing Marsh (1970) would be helpful. Additionally, in the surface area column in the raw data (Z00x density page), it appears that the first experimental run (moderate tolerance), there is a formula used to calculate the surface area that depends on the foil weight. However, for those data from the heat tolerant run, there is a number in the surface area column – no formula calculation. IF one copies the formula used to calculate surface area for the moderate tolerant: =(R2*100)/0.424. to the heat tolerant data, the calculated number is different. This is problematic.

I am interested in the fact that actual microplastic ingestion was not truly documented. I understand the photo the authors provided (Fig. 2) with the “before” (00:03) and “after” (00:13), with the one polyp associated with the pellet definitely changing shape from A to B implies ingestion. But, for example, the other pellet that disappeared from A to B, we just assume was NOT ingested because no polyp change was observed? I guess there is no normal way of quantifying actual ingestion by counting particles when water changes, or looking for particles when the polyps are processed? Indeed, the sentence (line 156) “…ingestion was an assumed response due to evidence…” is interesting. Might the lack of differences in microplastic effect here maybe be due to the fact that these corals are not really ingesting these microplastic particles much, and the assumption might be problematic?

Validity of the findings

I think, given some of the issues in the raw data, that perhaps a going over of the data analysis might be warranted. I would also perhaps mention somewhere in the discussion the above mentioned “assumption” of ingestion of plastics.

After addressing calculation concerns in the database, and perhaps the assumption of actual consumption of microplastics by these corals, that microplastics are not affecting this species is an interesting result. One always assumes that microplastics are bad. As the authors state in the introduction, this isn’t a foregone fact – there are equivocal results in the literature. The study does show experimental support for now just a 4 degree increase in water temperature negatively affects coral. Thais is a legitimate experimental result.

Reasons for their somewhat surprising findings are supported well. One might be interested in ideas about how to go about a deeper quantification of what is really happening here.

---

## Round 0.2 · Minor Revisions

Thank you for considering PeerJ and making the changes that were suggested by the reviewers.

A Section Editor notes that while the manuscript seems to be in order for the scientific merit, there remain fairly substantial edits to be made by the authors before it is ready for typesetting. A final round of minor revisions seems to be in order to ensure those corrections are incorporated into the published manuscript.

Reviewer 1 ·

Basic reporting

the improvement on the revised manuscript is quite evident. I have just attached some minor corrections at the revised file attached.

Experimental design

already corrected

Validity of the findings

already corrected

Additional comments

the improvement on the revised manuscript is quite evident. I have just attached some minor corrections at the revised file.

Annotated reviews are not available for download in order to protect the identity of reviewers who chose to remain anonymous.

Reviewer 2 ·

Basic reporting

Concerns have been addressed.

Experimental design

Concerns have been addressed

Validity of the findings

Concerns have been addressed

Additional comments

I am pleased with the thoughtful comments/responses to the issues that were raised previously on the manuscript. I feel they have adequately addressed my concerns.

---

## Round 0.3 · accepted · Accept

After all the revisions, I consider that the manuscript is accepted for publication.